# Maternal Gut Microbiome-Mediated Epigenetic Modifications in Cognitive Development and Impairments: A New Frontier for Therapeutic Innovation

**DOI:** 10.3390/nu16244355

**Published:** 2024-12-17

**Authors:** Shabnam Nohesara, Hamid Mostafavi Abdolmaleky, Faith Dickerson, Adrián A. Pinto-Tomás, Dilip V. Jeste, Sam Thiagalingam

**Affiliations:** 1Department of Medicine (Biomedical Genetics), Boston University Chobanian and Avedisian School of Medicine, Boston, MA 02218, USA; snohesar@bu.edu (S.N.); samthia@bu.edu (S.T.); 2Department of Surgery, Nutrition/Metabolism Laboratory, Beth Israel Deaconess Medical Center, Harvard Medical School, Boson, MA 02215, USA; 3Sheppard Pratt, Stanley Research Program, 6501 North Charles St., Baltimore, MD 21204, USA; fdickerson@sheppardpratt.org; 4Center for Research in Microscopic Structures and Biochemistry Department, School of Medicine, University of Costa Rica, San Jose 11501, Costa Rica; adrian.pinto@ucr.ac.cr; 5Global Research Network on Social Determinants of Mental Health and Exposomics, San Diego, CA 92037, USA; 6Department of Pathology & Laboratory Medicine, Boston University Chobanian and Avedisian School of Medicine, Boston, MA 02118, USA

**Keywords:** gut microbiota, cognition, gut–brain axis, histone modifications, DNA methylation, miRNAs

## Abstract

Cognitive impairment in various mental illnesses, particularly neuropsychiatric disorders, has adverse functional and clinical consequences. While genetic mutations and epigenetic dysregulations of several genes during embryonic and adult periods are linked to cognitive impairment in mental disorders, the composition and diversity of resident bacteria in the gastrointestinal tract—shaped by environmental factors—also influence the brain epigenome, affecting behavior and cognitive functions. Accordingly, many recent studies have provided evidence that human gut microbiota may offer a potential avenue for improving cognitive deficits. In this review, we provide an overview of the relationship between cognitive impairment, alterations in the gut microbiome, and epigenetic alterations during embryonic and adult periods. We examine how various factors beyond genetics—such as lifestyle, age, and maternal diet—impact the composition, diversity, and epigenetic functionality of the gut microbiome, consequently influencing cognitive performance. Additionally, we explore the potential of maternal gut microbiome signatures and epigenetic biomarkers for predicting cognitive impairment risk in older adults. This article also explores the potential roles of nutritional deficiencies in programming cognitive disorders during the perinatal period in offspring, as well as the promise of gut microbiome-targeted therapeutics with epigenetic effects to prevent or alleviate cognitive dysfunctions in infants, middle-aged adults, and older adults. Unsolved challenges of gut microbiome-targeted therapeutics in mitigating cognitive dysfunctions for translation into clinical practice are discussed, lastly.

## 1. Introduction

The number of individuals with cognitive impairment has increased in recent decades due to changes in lifestyle and the growing population of older adults worldwide [1,2]. The estimated global prevalence of cognitive impairment ranges from 5.1% to 41%, with a median of 19% [3]. A recent study analyzing data from 126 nationally representative surveys in 73 low- and middle-income countries found an overall prevalence of significant cognitive delay of 9.7% among 3- to 4-year-old children [4]. A meta-analysis study also reports that the global prevalence of mild cognitive impairment among community-dwelling persons aged 50 years and older is more than 15%, and it is influenced by education level, age, gender, and living area [5].

Due to the academic underachievement of affected children and adolescents and its economic burden on families and society, cognitive impairment is considered an important clinical target for preventive or therapeutic interventions [6]. Aside from genetic mutations (e.g., *APOE*, *APP*, *PSEN1*, *PSEN2*, *FMR1*, and *MECP2*) that are linked to the pathogenesis of major cognitive diseases, a key mechanism by which central nervous system (CNS) neurons contribute to the production and maintenance of behavioral memories at multiple levels involves epigenetic events [7,8]. Epimutations or abnormal changes to the epigenetic machinery associated with cognitive impairments are induced by various environmental factors, such as malnutrition, toxins, infections, and aberrations in the composition and diversity of microorganisms existing in the human gut [9], known as the gut microbiota, whose collective genomes constitute the microbiome [10]. The human body contains nearly 10 times more microbial cells than human cells, and the microbiome includes 3.3 million different genes—150 times more than those in the human genome [11]. The gut microbiome is involved in numerous physiological processes such as the development and modulation of the immune system and brain function, largely mediated by epigenetic modulations [12,13]. While overall health status is associated with optimal microbiome composition [14], an increased abundance of commensal bacteria modulates the host immune system, protecting it by inhibiting the colonization and invasion of pathogens [15,16]. A reduction in the abundance of commensal bacteria by risk factors such as the administration of antibiotics allows pathogens to accelerate the production of endotoxins and cause different types of diseases.

Gut dysbiosis due to major risk factors, such as aging and antibiotic consumption, can affect brain function and other physiological processes via substantial changes in the diversity and composition of the gut microbiome [17,18]. The “gut–brain axis” is responsible for bidirectional communications between the brain and the gastrointestinal (GI) tract via immune, neural, and endocrine pathways [19]. The gut microbiome is capable of influencing the enteric nervous system through neurotransmitters, such as serotonin and dopamine and its metabolites that affect the epigenome, in particular short-chain fatty acids (SCFAs) [20].

A growing body of evidence suggests that the resident bacteria of the GI tract also play a powerful role in the formation, processing, and storing of memories, which are generally described as cognitive functions in both health and disease states [21,22,23]. For example, the relative abundance of *actinobacteria* is associated with better cognitive test scores in obese individuals, and higher levels of *actinobacteria* are related to better cognitive performance [24]. In another study, better cognitive performance by participants from the Guangzhou Nutrition and Health Study has been linked to a higher abundance of *Bacteroides* in the GI tract [25].

Microbiome-derived metabolites are also capable of modulating host metabolism, brain function, and cognitive performance by acting as epigenetic regulators [9]. For example, in pre-clinical studies, butyrate and acetate act as histone deacetylase (HDAC) inhibitors and improve learning and memory [26]. Thus, the gut microbiome may be considered as a potential diagnostic or therapeutic target for cognitive decline. In the present review, we aim to summarize the current literature on the effects of the maternal gut microbiome, particularly during the embryonic period, on offspring cognitive impairments through epigenetic aberrations. Additionally, we explore the potential for gut microbiome-targeted strategies in the diagnosis and management of cognitive dysfunction and neurodegenerative diseases. It is worth noting that while there is some overlap between the present study and our recently published or submitted papers in the overall topic area related to Alzheimer’s Disease (AD) or microbiome-induced social behavior disarrays through epigenetic modification, this review focuses on the association between the maternal gut microbiome, its effects on the epigenome, and cognitive performance.

## 2. Methods

In this narrative review, our primary aim is to provide an overview of the correlations between alterations in the gut microbiome and epigenetic changes during embryonic and adult periods, and their impact on cognitive impairment. We collected relevant information by searching Scopus, PubMed, and Web of Science databases using the terms “microbiome” or “microbiota”, “cognitive” or “cognition”, along with “embryonic” or “maternal”, and terms related to specific epigenetic modifications, including DNA methylation, histone acetylation, and miRNAs. After excluding review papers, original research articles published from January 2010 through October 2024 were considered, with over 75 papers selected for further assessment. This review encompasses both clinical and pre-clinical studies, with a significant portion consisting of animal studies investigating the correlations between cognitive impairment, changes in gut microbiome composition, and epigenetic alterations.

## 3. Different Types of Factors Affecting Cognitive Performance by Alterations in Gut Microbiome

A wide range of factors such as nutrients, aging, antibiotic use, and environmental contaminants such as chemicals, air, and water pollution, as well as the state of sanitation, can heavily affect the functionality, structure, diversity, and composition of the gut microbiome and subsequently the epigenome and cognitive performance [27,28]. A summary of more recent studies linking different types of external or internal factors affecting cognitive performance by alterations in the gut microbiome is presented in Table 1.

The impact of various factors on cognitive performance through gut microbiome modifications is depicted in Figure 1.

## 4. Links Between Gut Microbial Pathology, Gut Metabolites Involved in Epigenetic Regulation, and Cognitive Impairment

In pre-clinical studies, cognitive decline during aging has been linked to abnormalities in the gut microbiome [44]. Transplantation of old mice gut microbiota to young mice induces inflammation, cognitive impairment, anxiety, and depression. These phenotypic alterations are associated with reduced mucin formation and elevated gut permeability, along with a decrease in the abundance of butyrate-producing bacteria and reduced expression of butyrate receptors, including FFAR2/3 (free fatty acid receptors 2 and 3) [45]. Fecal microbiota transplant from old donor mice also induces cognitive impairment, mediated by a decrease in SCFA-producing bacteria, particularly butyrate-producing bacteria such as *Ruminococcaceae*, *Faecalibaculum*, and *Lachnospiraceae* [46]. Likewise, in human studies, changes in the relative abundance of gut microbiota occur in patients with cognitive impairments such as AD and Parkinson’s disease compared to the control subjects. *Bacteroidetes*, *Firmicutes*, *Proteobacteria*, and *Actinobacteria* are the most prevalent phyla that show marked alterations in patients with cognitive impairments [47].

Cognitive decline during aging and in neuropsychiatric diseases is linked to epigenetic alterations caused by dysregulation of the gut microbiome as well. A total of 628 and 562 unique differentially methylated regions have been detected in “two human amyloid precursor protein knock-in mouse models” with cognitive decline associated with gut microbiome alterations [48]. In addition to DNA methylation changes, the gut microbiome produces epigenetically modifying metabolites such as butyrate and acetate, which act as HDAC inhibitors, thereby increasing histone acetylation levels. Therefore, these metabolites or their associated bacteria could be considered as potential epigenetic biomarkers for cognitive decline during aging and in neuropsychiatric diseases. In support of this idea, Fan et al. demonstrated a decline in the abundance of butyrate-producing bacteria, such as *Butyricimonas* and *Ruminococcus*, in patients with mild cognitive impairment versus control subjects [49]. Zhang et al. reported lower microbial diversity, reduced numbers of butyrate-producing bacteria (*Faecalibacterium* and *Ruminococcaceae*), and elevated levels of *Gammaproteobacteria* and *Proteobacteria*, known to be human pathogens, in individuals with mild cognitive impairment compared to healthy subjects [50]. McLeod et al. reported that cognitive scores in a group of predominately female African American adults with obesity were associated with the abundance of *Akkermansia muciniphila*, a bacterium involved in enriching butyrate-producing bacteria, enhancing gut barrier integrity, and suppressing inflammation [51].

Post-stroke cognitive impairment is also related to gut dysbiosis, with elevated levels of *Enterobacteriaceae* and lipopolysaccharide (LPS), and reduced levels of butyrate [52]. Moreover, plummeting levels of butyric acid, acetic acid, isobutyric acid, and propionic acid have been reported in the plasma of patients with diabetic cognitive impairment compared to type 2 diabetes mellitus (T2DM) patients without cognitive impairment [53]. Similarly, cognitive decline in congestive heart failure has been related to lower plasma concentrations of butyrate, propionate, and isovalerate [54]. Cognitive dysfunction in individuals with obesity is also associated with elevated levels of the phylum *Proteobacteria* and reduced levels of *Clostridium butyricum*, a butyrate-producing bacterial species [55]. In a prospective four-year follow-up study, Ma et al. showed that depletion of butyrate-producing bacterial species, associated with less frequent bowel movements and a higher abundance of pro-inflammatory bacterial species, is linked to lower cognitive performance in elderly humans (average age of 67 years) [56]. In another recent clinical study using free water imaging in subjects with mild cognitive impairment and AD compared to controls, Yamashiro et al. demonstrated that higher amounts of extracellular water in gray and white matter were strongly associated with neuroinflammation, gut dysbiosis, and a reduced abundance of butyrate-producing bacteria [57]. In an animal study, cognitive dysfunction caused by sleep deprivation in normal mice was also connected to decreased concentrations of butyrate, reduced abundance of *Lachnospiraceae*_NK4A136, and increased abundance of *Aeromonas* and LPS concentrations, which were reversed by butyrate and/or melatonin administration [58].

In addition to butyrate-producing bacterial species, changes in the gut acetate-producing bacteria can be considered as another potential biomarker associated with cognitive dysfunction. Elimination of acetate-producing bacteria and the subsequent long-period acetate deficiency, due to exposure to the non-absorbable antibiotic vancomycin, diminishes synaptophysin in the neurons of the hippocampus region, which in turn results in cognitive impairment in streptozotocin-induced type 1 diabetes mellitus mice [59]. In another study, Kong et al. [60] found reduced levels of butyrate/acetate-producing bacteria like *Acetobacter* and *Lactobacillus* in an AD model of *Drosophila* compared to controls. Liao et al. reported that cognitive impairment in sepsis-associated encephalopathy in mice was linked to drastic reductions in the levels of acetic acid and propionic acid and decreased abundance of SCFAs-producing bacteria, such as *Bacteroides*, *Bifidobacterium*, and *Allobaculum* [61]. Cognitive impairment caused by a high-salt diet in mice was also associated with reduced amounts of acetate and butyrate in fecal samples, disturbance in the blood–brain barrier, and increased levels of pro-inflammatory cytokines and apoptotic cell death in the cortex and hippocampus areas [34]. These studies collectively highlight the significant role of gut microbiome alterations in cognitive impairment across various conditions and age groups and study populations. The research consistently points to the importance of epigenetic mechanisms, in particular those mediated by SCFAs (such as butyrate and acetate) and their producing bacteria, in maintaining cognitive health. 

## 5. Relationship Between Maternal Prenatal Gut Microbiota and Offspring’s Cognitive Functions

The composition of the maternal prenatal gut microbiota is a key factor in a child’s social and cognitive development [62,63,64]. Maternal obesity contributes to a decrease in the abundance of *Ruminococcus*, *Bifidobacterium*, and *Blautia*, which is associated with impaired cognitive functioning in 36-month-old children [65]. As maternal obesity has been linked to children’s lower cognitive and social behavior capabilities [66,67], a high-fiber diet may reduce maternal obesity-induced social dysfunction and cognitive impairment in the offspring of female C57BL/6J mice through the modulation of bacterial composition and production of SCFAs [68]. Maternal dietary fiber intake is one example of a dietary approach that has shown efficacy in influencing the microglial transcriptome in both wild-type (WT) and 5xFAD (fed by a high-fat diet) mice and improving the offspring’s cognitive functions in 5xFAD mice by promoting the production of beneficial metabolites such as butyrate and acetate [69]. Moreover, it has been found that offspring cognitive impairment in mice and derangements of synaptic plasticity induced by maternal low-fiber diet can be reversed by butyrate intake via regulating the activity of histone deacetylase 4 (HDAC4) [70].

Nutritional deficiencies during early developmental stages may also result in cognitive impairment due to maternal immune activation induced by gut dysfunction in mice [71]. For example, maternal iron deficiency gives rise to spatial skill deficit in the offspring of Sprague Dawley (SD) rats, which is associated with alterations in the gut bacterial composition, including increased levels of the *Bacteroidaceae* genus *Bacteroides* and *Lachnospiraceae* genus *Marvinbryantia* and reduced plasma levels of acetate [72]. Besides iron involvement in gut microbial dysbiosis, there is an hypothesis that iron dysregulation may not only increase oxidative stress but also may cause reactivation of dormant microbes in the blood and other tissues such as the brain. In addition to iron dysregulation, the authors of this study propose that previous infections and dysbiosis also contribute to the reactivation of dormant microbes and an elevation in systemic inflammation through the production of neurotoxins, including pro-inflammatory LPS, which in turn increases blood–brain barrier permeability, neuroinflammation, cognitive dysfunction, and the risk of neuropsychiatric diseases [73].

Targeting the maternal gut microbiome and utilizing epigenetic profiling represent promising strategies for addressing neuropsychiatric disorders associated with cognitive impairment in human children and offspring in animal models. Butyrate-producing bacteria and plasma butyrate can be considered as potential markers to predict or detect cognitive dysfunction caused by various factors such as maternal diet. For example, the disruption of butyrate signaling by a maternal high-fructose diet (HFD) in rats could reduce plasma levels of butyrate and increase the levels of nuclear HDAC4 in the adult female offspring of rats. This was associated with the downregulation of GPR43 (a butyrate receptor) in the hippocampus. In the cultured astrocytes of the affected rats, these alterations could be reversed by butyrate treatment [74]. Furthermore, maternal low-fiber-diet-induced neurocognitive deficits are connected to reduced levels of butyrate in maternal serum and the brain tissue of the offspring in mice [70]. It has been reported that stress, both at the early and later stages of life, may cause cognitive dysfunction by decreasing neurotrophic factors (GDNF, NGF, and BDNF) that are strongly associated with reduced production of butyrate in rats [75]. An increase in anxiety-like behavior induced by maternal obesity is also associated with reduced levels of gut butyrate, but perinatal probiotic intake can promote resilience to neuropsychiatric diseases in the adult offspring of obese mice dams by enhancing gut butyrate and brain lactate levels [76].

Acetate is another gut microbiome metabolite with substantial epigenetic effects that may be used as a possible biomarker to assess the risk of cognitive impairment during pregnancy in human populations [77]. In fact, maternal high-fat-diet-induced cognitive and social–behavioral deficits in the offspring of mice are connected to reduced fecal levels of acetate and propionate; therefore, treatment with a mixture of acetate and propionate may reduce cognitive and social–behavioral dysfunctions [68]. In addition to these short-chain fatty acids, gut microbiota-derived vitamins such as B6 and B12, which are involved in DNA methylation processes, can be employed as markers to assess the risk of neurological diseases associated with cognitive decline in offspring. While the infant cognitive function at 40 days postnatal highly depends on maternal vitamin B12 status in the early stages of pregnancy [78], *Blautia* spp. has been found to be a master regulator of the biosynthesis of vitamin B12- and vitamin B12-related microbiome functions, which are highly correlated with the status of cognitive development in Black infants [79]. In addition to *Blautia*, non-pathogenic *Clostridium* species are involved in vitamin B12 biosynthesis and butyrate production, both of which are positively correlated with psychological resilience in primiparous mothers in the early postnatal period [80,81]. Interestingly, maternal plasma levels of folate, which contribute to DNA methylation machinery function, is another indicator of cognitive function in children [81,82]. Notably, folate and B12 deficiencies, as well as the composition of the gut microbiota, are also connected to cognitive dysfunction in animal models of AD, mediated by the disruption of hippocampal insulin signaling [83]. Therefore, the level of gut microbiota-derived metabolites can be indicators for the detection and utilization of therapeutic epi-drugs to prevent neurological diseases associated with cognitive deficits in infants.

In summary, maternal microbiome and epigenetic metabolite screening may serve as valuable tools for mitigating the impact of neuropsychiatric disorders. By identifying and addressing maternal microbial factors and the interlinked epigenetic modifications, mental health research teams may intervene proactively to promote favorable mental health outcomes for future generations.

## 6. Microbial and Dietary Interventions Contributing to Epigenetic Regulation for Cognitive Improvement

There are several types of gut microbiome-targeted therapeutics for improving cognitive impairments, as depicted in Figure 2. These preventive or therapeutic interventions may include fecal microbiota transplantation, prebiotics, probiotics, postbiotics, and dietary interventions. Several studies have shown the potential for gut microbiome-targeted therapeutics to prevent cognitive impairment in middle-aged and older adults. A case report of a patient with AD showed that fecal microbiota transplantation improved cognitive performance by increasing alpha diversity, restoring beneficial microbial populations and normalizing SCFA production [84]. 

Chen et al. reported that improving cognitive performance in patients with mild cognitive impairment was attributable to an increase in the abundance of butyrate-producing bacteria, notably *Prevotella* [85]. As another example, Su et al. found that chronic cerebral hypoperfusion causes cognitive impairment, which is associated with reduced concentrations of fecal acetic and propionic acid, as well as decreased levels of acetic acid in the hippocampus. While the exact mechanisms linking brain cerebral hypoperfusion and gut microbial alterations are unclear, fecal microbiota transplantation could prevent the reduction in acetic acid content and inhibit neuroinflammation by suppressing microglial and astrocytic activation and reducing oxidative phosphorylation impairment through mitochondrial metabolic reprogramming [86].

Probiotics may also play a role in mitigating cognitive decline associated with aging [87]. For instance, Aljumaah et al. found that the *Lactobacillus rhamnosus* GG probiotic could improve cognitive scores in middle-aged and older adults by reducing the abundance of bacteria in the genera *Dehalobacterium*, which has been linked to cognitive decline [88]. In another study, Zhu et al. reported that supplementation with *Bifidobacterium breve* HNXY26M4 could ameliorate cognitive impairments and neuroinflammation in an APP/PS1 mouse model of AD by increasing acetate and butyrate production [89]. Additionally, supplementation with multi-strain probiotics could enhance cognitive functions in acute stress circumstances by increasing the abundance of *Ruminococcaceae*_UCG-013 bacteria in the digestive tract [90].

The gut microbiome-derived metabolites categorized as postbiotics, such as short-chain fatty acids (SCFAs), especially butyrate, can exert anti-inflammatory and neuroprotective effects, helping to maintain normal cognitive function in older age through epigenetic mechanisms. Experimental evidence from animal models suggests that butyrate can improve obesity-induced cognitive impairments by preventing quinolinic acid-induced BDNF reductions through inducing H3K18ac (acetylation of lysine 18 of histone H3) at BDNF promoters [91]. Moreover, the improvement in cognitive impairments induced by Aβ25-35 in mice can be attributed to the enhancement of astro-glial mitochondrial function, promoting astrocyte differentiation into the A2-neuron-protective subtype, and improving the lactate shuttle between astrocytes and neurons via epigenetic mechanisms [92]. Due to its inhibition of histone deacetylase activity, butyrate can also alleviate aging-associated recognition memory deficits in rodent models [93].

Specialized diets, such as those rich in prebiotics that promote the secretion of SCFAs, particularly butyrate, can enhance cognitive function. For instance, a modified Mediterranean–ketogenic diet may improve cognition in individuals with mild cognitive impairment by increasing the levels of butyrate and propionate [94]. Similarly, restriction of methionine, an amino acid mainly found in animal food, can alleviate high-fat-diet (HFD)-induced cognitive decline by significantly elevating acetate and butyrate levels and SCFA-producing bacteria in mice [95]. The intake of soluble fiber can also ease cognitive impairment in a 6-month-old male APP/PS1 mouse model of AD by enhancing butyrate levels and suppressing astrocyte activation [96].

Plant-based diets are also adequate for diminishing cognitive decline by increasing the relative abundance of SCFA-producing bacteria. For instance, a 12-week treatment with verbascoside, an ingredient of the *Herba Cistanche* plant, could attenuate cognitive impairments in db/db mice by improving gut dysbiosis, enhancing gut microbiota diversity, decreasing the abundance of *Escherichia–Shigella*, and increasing the abundance of butyrate-producing bacteria such as *Roseburia* and *Intestinimonas*, as well as other beneficial species including *Alistipes* [97]. Tian et al. found that the polysaccharides from the fruits of *Lycium barbarum* have a capacity for reducing high-fat- and high-fructose-diet-induced cognitive deficits by increasing the contents of n-butyric, i-butyric, n-valeric, i-valeric, and propionic acids and overexpression of GPR43, GPR41, and GPR109A [33].

Consumption of specific proteins is another possible remedy for alleviating cognitive impairments through the elevation of SCFA-producing bacteria. For example, lactoferrin has been found to improve Western diet-induced cognitive dysfunction and neuroinflammation in mice by elevating the levels of *Roseburia* (a butyrate-producing bacterium) and *Bacteroidetes*, while enhancing the expression of tight junction proteins [98].

## 7. Conclusions and Future Perspectives

A growing body of evidence supports the notion that the gut microbiome is involved in the modulation of immune responses, brain plasticity, and cognitive performance via the gut–brain axis. A deeper understanding of the links between the gut microbiome, cognitive function, and epigenetic mechanisms may pave the way for the development of promising preventive or therapeutic strategies for cognitive impairments via modulation of the microbiota–gut–brain axis through epigenetic alterations. However, before this promise becomes a reality, the clinical application of gut microbiome-targeted therapeutics to alleviate cognitive impairments must overcome several challenges. For instance, it is currently challenging to draw clear conclusions about the relationship between shifts in gut microbiota composition and cognitive impairment. This is because researchers have focused on different taxonomic levels, with some investigating differences at the genus level, others at the phylum level, and some at the species level. Adopting a uniform taxonomic scale would be beneficial for comparing the relationship between specific microbial taxa and cognitive functions across different age groups. Moreover, the effects of certain bacterial species, such as *Bacteroides*, on cognitive function remain unclear, with conflicting findings reported in different studies [99,100]. To address these controversies, it is suggested that future research should compare the relative abundance of bacteria in cognitive dysfunction across different age groups and disease states (such as Alzheimer’s disease or diabetes mellitus). This approach would help to identify the effects of potential confounding variables and provide a more comprehensive understanding of the microbiome–cognition relationship.

As another example, translating findings from animal studies to humans presents significant challenges due to the notable differences between the human and rodent microbiota–gut–brain axes, particularly regarding variations in brain structure, especially in the prefrontal cortex and the frontoinsular area. One additional challenge is that a large number of recent studies have not investigated the long-term effects of therapeutic strategies such as probiotics or fecal microbiota transplantation in human populations. Another point to consider is the need to determine more precisely the optimal dosage and timing of dietary or probiotic interventions in clinical studies to minimize potential side effects in patients. An additional consideration is selecting the human populations most likely to benefit, such as pregnant women, infants, older adults, etc. The small sample size of patients is among other major limitations of existing studies, potentially affecting the reliability of current findings. Other challenges include the high heterogeneity of populations with cognitive impairment and the lack of control for various factors that influence gut microbiome composition, such as medications, age, disease comorbidities, seasonal factors, and nutritional habits. These limitations underscore the need for larger, more controlled studies to establish more robust associations between gut microbiome alterations and cognitive impairment. The high degree of heterogeneity among populations with cognitive impairment suggests that lifestyle, local conditions, and diet may exert a strong impact on both gut microbiome composition and cognitive functioning. Considering these limitations, further prospective studies should be implemented to clarify ambiguities and improve our understanding of whether gut microbiota and their derived metabolites could be employed as potential biomarkers or preventive and therapeutic tools for cognitive disorders via epigenetic mechanisms. Additionally, given the controlled laboratory settings in animal studies, more investigations are necessary to clarify the impact of confounding variables in humans. These studies should also explore the potential role of these variables in determining the composition of the maternal gut microbiome and their effects on offspring cognitive function via epigenetic mechanisms.

## Figures and Tables

**Figure 1 nutrients-16-04355-f001:**
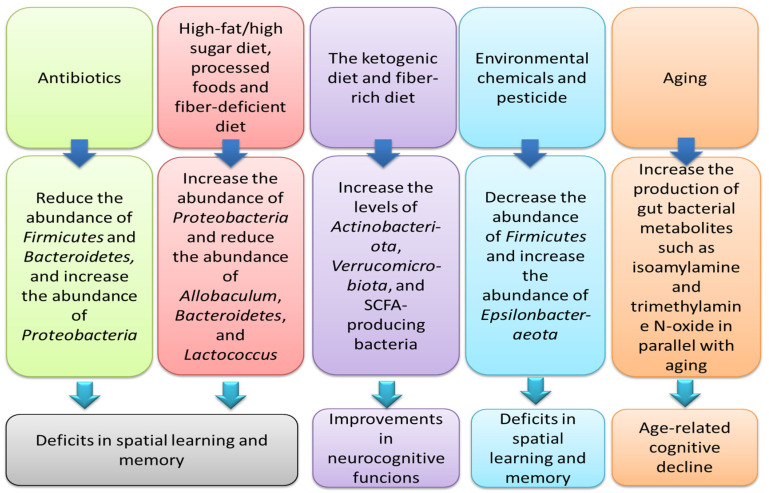
Association between various factors (nutritional interventions, age, antibiotics, and environmental factors such as chemicals), changes in the composition of the gut microbiome, and cognitive performance.

**Figure 2 nutrients-16-04355-f002:**
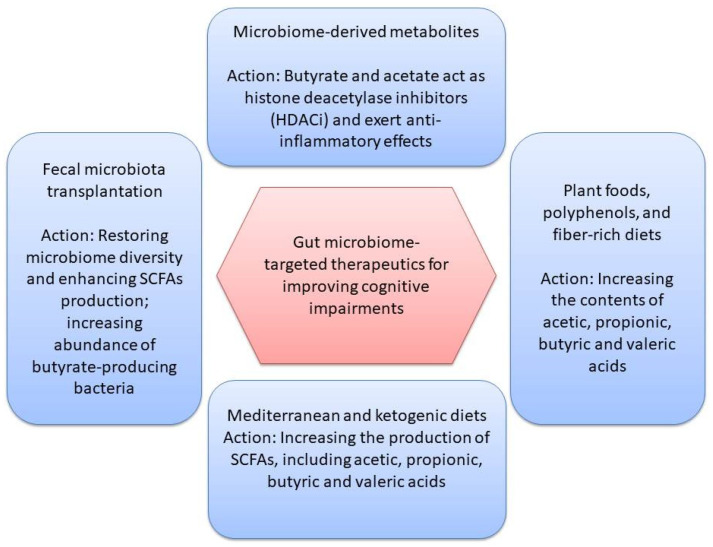
Gut microbiome-targeted therapeutics for improving cognitive impairments.

**Table 1 nutrients-16-04355-t001:** Different factors that alter gut microbiota composition and cognitive performance.

Factor/Type of Study	Effect on Microbiome	Impact on Cognitive Performance	Reference
Antibiotic (a cocktail including vancomycin, neomycin, ampicillin, metronidazole, and amphotericin-B)/in Aβ1–42-treated mice	Antibiotic-induced gut microbiota depletion	Exacerbation of cognitive deficits after treatment with antibiotic cocktail	[29]
Antibiotic/in human	Derangements in the usual composition of gut microbial communities due to neonatal exposure to antibiotics	Changes in discrimination responses and auditory processing at 1 month of age in infants exposed to antibiotics	[30]
Antibiotic (a cocktail, including vancomycin, imipenem, bacitracin, neomycin, and amphotericin B)/in mice	Reduced abundance of *Firmicutes* and *Bacteroidetes*, but increased abundance of *Proteobacteria*	Association between gut microbiota depletion by antibiotics and impairments in memory and cognitive functions	[31]
Long-term consumption of a high-fat diet/in male Wistar rats	Gut dysbiosis and systemic inflammation	Elevated amyloid-β in the brain and cognitive decline after 40 weeks of high-fat diet utilization	[32]
High-fat and high-fructose diet/in mice	Increased abundance of *Proteobacteria* and reduced abundance of *Allobaculum* and *Lactococcus*	Cognitive impairment	[33]
High-salt diet/in mice	Reduced concentrations of butyrate, acetate, and propionate	Impaired learning and memory capacities after 8 weeks	[34]
Ketogenic diet and hypoxia/in mice	Microbiota-mediated cognitive impairment	Association between *Bilophila wadsworthia* and cognitive disturbances	[35]
Ketogenic diet/in rats with pilocarpine-induced status epilepticus	Microbiota-mediated cognitive impairment	Association between higher *Actinobacteriota* and *Verrucomicrobiota* levels after a ketogenic diet intake and improvements in learning and memory	[36]
A fiber-deprived diet/in mice	Reduced abundance of *Bacteroidetes* and elevated abundance of *Proteobacteria*	Impairments in the cognitive functions	[37]
Dietary advanced lipoxidation end-products present in ultra-processed, heat-processed, and fat-enriched foods/in mice	Increased abundance of *Muribaculum* and *Parasutterella* and decreased abundance of *Faecalibaculum* and an unclassified *Bacteroidales*	Deficits in cognition functions	[38]
Age/in old mice	Elevated levels of TMAO (trimethylamine *N*-oxide), a gut microbiome-derived metabolite, in parallel with aging	Association between TMAO and cognitive decline in aging	[39]
Age/in human	An association between higher abundance of the bacterial family *Carnobacteriaceae* and improved episodic secondary memory; a greater abundance of *Clostridiaceae* is linked to better continuity of attention	Reductions in cognitive performance in older Australians	[40]
Age/in mice	Increased amounts of isoamylamine, a gut bacterial metabolite in parallel with aging	Association between elevated levels of isoamylamine and age-related cognitive decline	[41]
Environmental chemicals (exposure to inorganic arsenic and fluoride)/in rats	Reduced *Firmicutes*, but increased abundance of *Epsilonbacteraeota* and *Bacteroidetes* in animals exposed to arsenic	Deficits in spatial learning and memory	[42]
Pesticide (exposure to Tebuconazole)/in mice	Disturbances in the *Firmicutes/Bacteroidetes* ratio, systemic immune factors, and production of neurotransmitters	Derangements in synaptic function integrity, memory, and spatial learning	[43]

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
