# Peer review of "Maternal Gut Microbiome-Mediated Epigenetic Modifications in Cognitive Development and Impairments: A New Frontier for Therapeutic Innovation"

_nutrients, 2024, doi:10.3390/nu16244355_

Round 1

Reviewer 1 Report

Comments and Suggestions for Authors

This article explores the impact of the gut microbiome on cognitive function through epigenetic regulation and highlights the bidirectional mechanism of action of the gut-brain axis in health and disease. Studies have shown that changes in the diversity and composition of the gut microbiota are strongly associated with cognitive dysfunction, especially under risk factors such as aging, antibiotic use, and poor diet. Microbial metabolites such as short-chain fatty acids (such as butyric acid and acetic acid) play a key role in regulating gene expression, neuroinflammation, and neurocognitive function. In addition, the profound influence of maternal gut microbial composition on the cognitive development of offspring has also been analyzed in detail. These findings provide a theoretical basis for targeting and regulating gut microbes to improve cognitive function through dietary intervention, probiotics, and fecal transplantation. Overall, the article systematically summarizes the potential diagnostic and therapeutic value of gut microbes in cognitive health, but more high-quality clinical trials are needed to validate their efficacy to advance the practical application of relevant intervention strategies.

1. The author lacks logic in the language description of the full text, and suggests sorting out the logic.

2. In terms of the cited sources of the data, the paper cites a large number of data and research results, such as "the global prevalence of cognitive impairment is 5.1%-41%", but lacks a critical analysis of the data sources to clarify its reliability and applicability. It is suggested to add more explanations on the credibility of the data.

3. When referring to bacterial genera in different studies, such as Ruminococcin and Bacteroides, their roles sometimes seem to contradict each other. Some studies have linked it to cognitive improvement, while others have linked it to cognitive decline.

4. Limitations of the results are less discussed, such as whether there are unresolved confounding factors.

5. It is suggested that the author add suggestions for future research directions.

Comments on the Quality of English Language

none

Author Response

Reviewer 1

Comments and Suggestions for Authors

This article explores the impact of the gut microbiome on cognitive function through epigenetic regulation and highlights the bidirectional mechanism of action of the gut-brain axis in health and disease. Studies have shown that changes in the diversity and composition of the gut microbiota are strongly associated with cognitive dysfunction, especially under risk factors such as aging, antibiotic use, and poor diet. Microbial metabolites such as short-chain fatty acids (such as butyric acid and acetic acid) play a key role in regulating gene expression, neuroinflammation, and neurocognitive function. In addition, the profound influence of maternal gut microbial composition on the cognitive development of offspring has also been analyzed in detail. These findings provide a theoretical basis for targeting and regulating gut microbes to improve cognitive function through dietary intervention, probiotics, and fecal transplantation. Overall, the article systematically summarizes the potential diagnostic and therapeutic value of gut microbes in cognitive health, but more high-quality clinical trials are needed to validate their efficacy to advance the practical application of relevant intervention strategies.

We sincerely appreciate your valuable comments. We have carefully revised the manuscript in accordance with your suggestions. Appropriate amendments have been made, and additional information has been included, all of which are highlighted in green for your convenience.

  1. The author lacks logic in the language description of the full text, and suggests sorting out the logic.

Author response and action taken: We implemented the comment in the revised version as follows:

We have modified text (highlighted in green on page 2, line 52-54, lines 93-98) and added new sentences on page 7, lines 201-206, in response to this comment. The first three sentences of Methods (new section) also address this comment.

  1. In terms of the cited sources of the data, the paper cites a large number of data and research results, such as "the global prevalence of cognitive impairment is 5.1%-41%", but lacks a critical analysis of the data sources to clarify its reliability and applicability. It is suggested to add more explanations on the credibility of the data.\

Author response and action taken: We have added more studies and explanations to enhance the credibility of the data, highlighted in green in the introduction, page 2, lines 46-54. 

  1. When referring to bacterial genera in different studies, such as Ruminococcin and Bacteroides, their roles sometimes seem to contradict each other. Some studies have linked it to cognitive improvement, while others have linked it to cognitive decline.

Author response and action taken: we added it as a limitation of existing literature in conclusion and future perspective section, page 10, lines 368 and 369 (before references 96,97).  

  1. Limitations of the results are less discussed, such as whether there are unresolved confounding factors.

Author response and action taken: we added more information about limitations of the results and unresolved confounding factors. Please see green highlights in conclusion and future perspective section, pages 10 and 11, lines 361-374 and lines 384-393.    

  1. It is suggested that the author add suggestions for future research directions.

Author response and action taken: we added suggestions for future research directions. Please see green highlights in conclusion and future perspective section. Lines 384-402, in particular lines 393-402 are related to this comment.   

Reviewer 1

Comments and Suggestions for Authors

This article explores the impact of the gut microbiome on cognitive function through epigenetic regulation and highlights the bidirectional mechanism of action of the gut-brain axis in health and disease. Studies have shown that changes in the diversity and composition of the gut microbiota are strongly associated with cognitive dysfunction, especially under risk factors such as aging, antibiotic use, and poor diet. Microbial metabolites such as short-chain fatty acids (such as butyric acid and acetic acid) play a key role in regulating gene expression, neuroinflammation, and neurocognitive function. In addition, the profound influence of maternal gut microbial composition on the cognitive development of offspring has also been analyzed in detail. These findings provide a theoretical basis for targeting and regulating gut microbes to improve cognitive function through dietary intervention, probiotics, and fecal transplantation. Overall, the article systematically summarizes the potential diagnostic and therapeutic value of gut microbes in cognitive health, but more high-quality clinical trials are needed to validate their efficacy to advance the practical application of relevant intervention strategies.

We sincerely appreciate your valuable comments. We have carefully revised the manuscript in accordance with your suggestions. Appropriate amendments have been made, and additional information has been included, all of which are highlighted in green for your convenience.

  1. The author lacks logic in the language description of the full text, and suggests sorting out the logic.

Author response and action taken: We implemented the comment in the revised version as follows:

We have modified text (highlighted in green on page 2, line 52-54, lines 93-98) and added new sentences on page 7, lines 201-206, in response to this comment. The first three sentences of Methods (new section) also address this comment.

  1. In terms of the cited sources of the data, the paper cites a large number of data and research results, such as "the global prevalence of cognitive impairment is 5.1%-41%", but lacks a critical analysis of the data sources to clarify its reliability and applicability. It is suggested to add more explanations on the credibility of the data.\

Author response and action taken: We have added more studies and explanations to enhance the credibility of the data, highlighted in green in the introduction, page 2, lines 46-54. 

  1. When referring to bacterial genera in different studies, such as Ruminococcin and Bacteroides, their roles sometimes seem to contradict each other. Some studies have linked it to cognitive improvement, while others have linked it to cognitive decline.

Author response and action taken: we added it as a limitation of existing literature in conclusion and future perspective section, page 10, lines 368 and 369 (before references 96,97).  

  1. Limitations of the results are less discussed, such as whether there are unresolved confounding factors.

Author response and action taken: we added more information about limitations of the results and unresolved confounding factors. Please see green highlights in conclusion and future perspective section, pages 10 and 11, lines 361-374 and lines 384-393.    

  1. It is suggested that the author add suggestions for future research directions.

Author response and action taken: we added suggestions for future research directions. Please see green highlights in conclusion and future perspective section. Lines 384-402, in particular lines 393-402 are related to this comment.   

Reviewer 2 Report

Comments and Suggestions for Authors

The review entitled “Maternal Gut Microbiome Mediated Epigenetic Modifications in Cognitive Development and Impairments: A New Frontier for Therapeutic Innovation” focuses on the association between the maternal gut microbiome, its effects on the epigenome and cognitive performance, emphasizing the relationship between cognitive decline, alterations in the gut microbiome and epigenetic alterations during the embryonic and adult periods. It explores, among other issues, the potential of maternal gut microbiome signatures as epigenetic biomarkers to predict the risk of cognitive decline in older adults, providing information on important aspects to be taken into account for the development of therapeutic strategies for cognitive decline through the modulation of the gut-brain axis of the microbiota by epigenetic alterations and the challenges that need to be overcome. It is therefore interesting information. The introduction and development of the topic is well organized and with adequate references. However, it requires further review of some aspects:

The authors must indicate the methodology followed in the aspects that must be taken into account for a review of this type.

Method:

- Sources

- Types of works

- Temporality (years)

- Databases including those of clinical trials

- Descriptors used

- Total number of works found

- Conditions established for the final selection of the works used

Author Response

Reviewer 2

Comments and Suggestions for Authors

The review entitled “Maternal Gut Microbiome Mediated Epigenetic Modifications in Cognitive Development and Impairments: A New Frontier for Therapeutic Innovation” focuses on the association between the maternal gut microbiome, its effects on the epigenome and cognitive performance, emphasizing the relationship between cognitive decline, alterations in the gut microbiome and epigenetic alterations during the embryonic and adult periods. It explores, among other issues, the potential of maternal gut microbiome signatures as epigenetic biomarkers to predict the risk of cognitive decline in older adults, providing information on important aspects to be taken into account for the development of therapeutic strategies for cognitive decline through the modulation of the gut-brain axis of the microbiota by epigenetic alterations and the challenges that need to be overcome. It is therefore interesting information. The introduction and development of the topic is well organized and with adequate references. However, it requires further review of some aspects:

We highly appreciate your feedback on our manuscript. We have revised the manuscript based on your valuable comments.

The authors must indicate the methodology followed in the aspects that must be taken into account for a review of this type.

Author response: Relevant information has been added to the manuscript in response to this comment, highlighted in blue under Methods, page 3, lines 105-116.  

Method:

- Sources

- Types of works

- Temporality (years)

- Databases including those of clinical trials

- Descriptors used

- Total number of works found

- Conditions established for the final selection of the works used

Round 2

Reviewer 2 Report

Comments and Suggestions for Authors

The authors have made the appropriate changes, improving the work considerably, so it can be published in the current version.